# High resolution monitoring of marine protists based on an observation strategy integrating automated on board ship filtration and molecular analyses

Katja Metfies[1], Friedhelm Schroeder[2], Johanna Hessel[1], Jochen Wollschläger[2], Sebastian Micheller[1], Christian Wolf[1], Estelle Kilias[1], Pim Sprong[1], Stefan Neuhaus[3]; Stephan Frickenhaus[3] Wilhelm Petersen[2]

[1]Helmholtz Young Investigators Group PLANKTOSENS, Alfred Wegener Institute Helmholtz Centre for Polar and Marine Research, Bremerhaven, D-27570, Bremerhaven, Germany

[2]*In-situ* Measuring Systems, Helmholtz Zentrum Geesthacht Centre for Materials and Coastal Research, Geesthacht, D-21502, Germany

[3]Scientific Computing, Alfred Wegener Institute Helmholtz Centre for Polar and Marine Research, Bremerhaven, D-27570, Bremerhaven, Germany

*Correspondence to*: Katja Metfies (Katja.Metfies@awi.de)

**Abstract**
Information on recent biomass distribution and biogeography of photosynthetic marine protists with adequate
temporal and spatial resolution is urgently needed to better understand consequences of environmental change
for marine ecosystems. Here we introduce and review a molecular-based observation strategy for high resolution
assessment of these protists in space and time. It is the result of extensive technology developments, adaptations
and evaluations which are documented in a number of different publications and the results of recently
accomplished field testing, which are introduced in this review. The observation strategy is organized at four
different levels. At level 1, samples are collected at high spatio-temporal resolution using the remote-controlled
automated filtration system AUTOFIM. Resulting samples can either be preserved for later laboratory analyses,
or directly subjected to molecular surveillance of key species aboard the ship via an automated biosensor system
or quantitative polymerase chain reaction (level 2). Preserved samples are analyzed at the next observational
levels in the laboratory (level 3 and 4). This involves at level 3 molecular fingerprinting methods for a quick and
reliable overview of differences in protist community composition. Finally, selected samples can be used to
generate a detailed analysis of taxonomic protist composition via the latest Next Generation Sequencing
Technology (NGS) at level 4. An overall integrated dataset of the results based on the different analyses provides
comprehensive information on the diversity and biogeography of protists, including all related size classes. At
the same time the cost effort of the observation is optimized in respect to analysis effort and time.

**Keywords**
Molecular observation strategy, Marine protists, Next Generation Sequencing, Automated Sampling, Molecular
fingerprinting, Quantitative PCR










## 1 Introduction

It is expected that marine ecosystems will be affected by climate change in multiple ways, including rising atmospheric $CO_2$ levels, shifts in temperature, circulation, stratification, nutrient input, oxygen content, and ocean acidification. In summary, these changes will strongly impact marine biota and ecosystems with consequences for abundance, diversity, spatial distribution, biogeography, or dominance of marine species (Doney et al., 2012). Marine plankton, comprising prokaryotic and eukaryotic microbes (bacteria and protists) as well as small or juvenile metazoans, is of utmost importance for the functioning of marine ecosystems. It is traditionally divided by its size into three classes: The microplankton (20-200 μm), the nanoplankton (20-2 μm), and the picoplankton (<2 μm). Within these groups of organisms, phytoplankton as the photosynthetic active part of the plankton accounts for roughly half of global net primary productivity (NPP) (Field et al., 1998) and is fundamental for any marine ecosystem function or service. As a consequence, changes in phytoplankton community structures and biogeography as a response to climate change are currently topical issues in marine ecology. Moreover, marine phytoplankton is very well suited to serve as an indicator of climate change (Nehring, 1998), because its dynamics are closely coupled to environmental conditions (Acevedo-Trejos et al., 2014). Despite the necessity and advantage of using marine phytoplankton to assess consequences of climate change, the task is also challenging in various ways. Marine phytoplankton distribution displays high spatial heterogeneity or "patchiness" (Mackas et al., 1985) and a pronounced seasonality as a consequence of physical and chemical oceanographic processes (Boersma et al., 2016; Bresnan et al., 2015). Furthermore, there are difficulties with the taxonomic surveillance of species in the pico-, or nano-fraction, related to their cell size and insufficient morphological features (e.g. Caron et al. 1999). As a consequence it is very challenging to provide information on composition, occurrence, and dynamics of phytoplankton with adequate spatial and temporal resolution. Together with the difficulties to financially support and maintain long time series these challenges might account for the relatively small number of marine phytoplankton long-term time series worldwide. Among them, one long lasting time series, the Helgoland Roads Time Series, is maintained by the Alfred Wegener Institute Helmholtz Centre for Polar- and Marine Research at the island Helgoland in the German Bight (North Sea). The dataset comprises information on abundance of phytoplankton on a daily basis since 1962 (Kraberg et al., 2015; Wiltshire et al., 2009). However, it does not provide information on the abundance of the smallest phytoplankton species and is restricted to one sampling point. The latter restriction is overcome by a second major long term marine observation programme that is operated by the Sir Alistair Hardy Foundation for Ocean Science in Plymouth, UK: the **C**ontinuous **P**lankton **R**ecorder (CPR) Survey's marine observation programme (McQuatters-Gollop et al., 2015). Together with its sister surveys it provides large-scale information on marine plankton distribution, mainly in the North Atlantic and the North Sea since the first surveys in 1931. Unfortunately, the CPR-approach is restricted to zooplankton and larger phytoplankton e.g. diatoms. Again, the ecological relevant pico-phytoplankton fraction is omitted. However, the smaller phytoplankton is to a certain degree included in the surveys of the FerryBox project implemented by the Helmholtz Centre Geesthacht in the North Sea. A FerryBox is an autonomous device located on ships of opportunity that has the capability to autonomously generate information on the plankton composition and a number of other parameters for the North Sea (Petersen, 2014). Here, phytoplankton is characterized on the basis of the pigment composition present, which is estimated via multi-channel fluorescence measurements. All phytoplankton groups and size fractions are included in this analysis, but this approach is only suited for the identification of larger taxonomic algal

groups. Furthermore, spectrally similar groups (e.g. diatoms and dinoflagellates) cannot be distinguished by this
method. Thus the FerryBox project lacks information on species composition of phytoplankton.
Overall, these long term monitoring programmes and other current marine plankton observation approaches
have already given important information and indication on climate related change in the marine plankton
community. Nevertheless, each of them is limited in one or the other way: (i) the ongoing long time series are
mainly limited to one or small numbers of sampling points; (ii) they do not provide a holistic view of changes at
the base of marine food webs, because they neglect the pico- and most of the nano-phytoplankton; (iii) broad
taxonomic knowledge is required for the identification of taxa at species level; (iv) fluorescent characterization
of phytoplankton is restricted to the identification of larger taxonomic groups; (v) they are costly if larger
numbers of samples need to be processed. To address these shortcomings and challenges of current observation
approaches, it is of utmost importance to develop efficient automated high throughput approaches and
observation strategies that allow reliable surveillance of all phytoplankton size classes with adequate spatio-
temporal resolution. Over the past decade numerous publications demonstrated the power of molecular methods
for the observation of marine plankton organisms, especially of those that are missing distinct morphological
features (Metfies et al., 2010; Wolf et al., 2014a; Wollschlaeger et al., 2014). Previous publications have shown
the power of the analysis of ribosomal genes (rRNA-genes) to gain new insights into the phylogeny and
biogeography of prokaryotic and eukaryotic micro-organisms (Comeau et al., 2011; Sunagawa et al., 2015). The
genes coding for the rRNA are particularly well suited for phylogenetic analysis and taxonomical identification,
because they are universally present in all cellular organisms. Furthermore, rRNA genes are of relatively large
size and contain both highly conserved and variable regions with no evidence for lateral gene transfer (Woese,
1987). The continually growing number of available algal 18S rDNA-sequences, e.g. in the Ribosomal Database
Project (Quast et al., 2013), and phylogenetic analysis makes it possible to design hierarchical sets of probes that
specifically target the 18S-rDNA of different taxa (Metfies and Medlin, 2007; Thiele et al., 2014). The probes
can be used in combination with a wide variety of hybridization based methods, such as RNA-based nucleic acid
biosensors (Diercks et al., 2008a; Ussler et al., 2013) quantitative PCR (Bowers et al., 2010; Toebe et al., 2013)
or fluorescence *in situ* hybridization (FISH) (Thiele et al., 2014) to identify marine microbes. Other methods,
such as molecular fingerprinting approaches and Next Generation Sequencing provide information on variability
and composition of whole microbial communities. The molecular fingerprinting method **A**utomated **R**ibosomal
**I**ntergenic **S**pacer **A**nalysis (ARISA) is a quick, cost-effective and meaningful method to determine overall
variability in phytoplankton community composition (Kilias et al., 2015) that is independent of the size or
morphology of target organisms. In contrast, Next Generation Sequencing (NGS) of ribosomal genes allows high
resolution, taxon-specific assessments of protist communities, including their smallest size fractions and the rare
biosphere (de Vargas et al., 2015; Kilias, 2014).
Here, we introduce a combined molecular-based observation strategy that allows observation of current
phytoplankton composition, distribution, and dynamics at adequate spatial and temporal scales. The resulting
data sets can be used to estimate possible alterations related to climate or environmental change. Our strategy is
the result of technical developments and the integration of latest sampling- and molecular tools in an advanced
molecular-based observation approach that will optimize marine microbial observation in general, while
phytoplankton was in the focus of our developments. In the future our molecular observation strategy is intended
to cut down surveillance costs and provide information on marine microbial biodiversity with unprecedented
resolution. It is a development of the Helmholtz Young Investigators Group PLANKTOSENS (Assessing
Climate Related Variability and Change of Planktonic Foodwebs in Polar Regions and the North Sea) carried out
within the framework of COSYNA (Coastal Observing System for Northern and Arctic Seas). Here, we review
major published results that lead to the development of the molecular observation strategy and demonstrate the
applicability of newly developed sampling technology within the observation strategy. Special emphasis was put
on observation of Arctic pico-phytoplankton that constitutes a major contribution to pelagic Chl $a$ biomass
during summer (Metfies et al., 2016).
**2 Material and Methods**
**2.1 Sampling**
Water samples analyzed in this study were collected during expeditions PS85 (June 2014) and PS96 (May/June
2015) of RV Polarstern to the Arctic Ocean. Samples from deeper water layers containing the deep chlorophyll
maximum (DCM) were taken with a rosette sampler equipped with 24 Niskin bottles (12 L per bottle) and
sensors for Chl $a$ fluorescence, temperature and salinity (CTD). Samples collected via CTD were taken during
the up-casts at the vertical maximum of Chl $a$ fluorescence determined during the down-casts. The sampling
depths varied between 10–50 m. Two litres of water subsamples were taken in PVC bottles from the Niskins.
Particulate organic matter for molecular analyses was collected by sequential filtration of one water sample
through three different mesh sizes (10 μm, 3 μm, 0.4 μm) on 45 mm diameter Isopore Membrane Filters at 200
mbar using a Millipore Sterifil filtration system (Millipore, USA). Subsequent to sampling the filters were stored
at -20°C until further analyses.
Additional samples were collected from a depth of ~ 10 m with the **Auto**mated **fi**ltration device for
**m**arine microorganisms (AUTOFIM), which is coupled to the ship's pump system. Fitting and programming of
the device does not require special expertise if it is done according to the manufacturer's protocol. All steps
related to the filtration process, including application of Lysis Buffer RLT (Qiagen, Germany), were carried out
automatically by AUTOFIM. In this study, two liters of sea water were collected and filtrated on a filter with 0.4
μm pore size at 200 mbar. Subsequent to filtration, particulate organic matter on the filter was re-suspended with
600 μl Lysis Buffer RLT (Qiagen, Germany) and stored at -80°C until further processing in the laboratory. The
filtration device was cleaned after each filtration step by rinsing the device with fresh-water.
**2.2 Environmental parameters**
Standard oceanographic parameters (salinity, temperature, Chl $a$ fluorescence, turbidity, chromophoric dissolved
organic matter, dissolved oxygen, pH, nutrients) were measured at the sampling sites by the FerryBox-System
(Petersen, 2014) deployed on board RV Polarstern. The measurement interval was 1 min, and the water intake of
the system was identical to the water supply of AUTOFIM. To prevent biofouling of the sensors, the FerryBox
performed a cleaning cycle including an acid wash and freshwater rinsing once per day.
**2.3 DNA isolation**
Isolation of genomic DNA from the field samples was carried out using the E.Z.N.A TM SP Plant DNA Kit Dry
Specimen Protocol (Omega Bio-Tek, USA) following the manufacturer's protocol. The resulting DNA-extracts
were stored at -20 °C.

**2.4 DNA quality**


The integrity of the genomic DNA isolated from water samples collected with AUTOFIM was assessed using
the Agilent DNA 7500 kit (Agilent Technologies, USA) according to the manufacturer's protocol. A volume of
1µl DNA was applied to the flow cell.

**2.5 ARISA**


PCR-amplification and subsequent determination of the size of the PCR fragments, and statistical analyses
related to ARISA were accomplished as described previously in the studies contributing to the development of
the molecular observation strategy (e.g. Kilias et al., 2015). This included the determination of variability in the
length of the internal transcribed spacer 1 (ITS1) amplified via a specific primer set from genomic DNA
extracted from field samples.

**2.6 454-Pyrosequencing**


Sequencing of protist communities via 454-pyrosequencing was based in all studies reviewed in this manuscript
on amplification of a ~ 670 bp fragment of the 18S rDNA containing the hypervariable V4 region. Sequence
library preparation and data analysis was described previously in the studies contributing to the development of
the molecular observation strategy ( Kilias et al., 2013; Metfies et al., 2016). Thus, for more detailed
information, the reader is referred to these publications

**2.7 Quantitative PCR-assay**


The quantitative PCR was carried out in a nested two-step approach. We used this nested approach, because it
minimized the variability between technical replicates of q-PCR data obtained from analyses of field samples.
The applicability of the nested approach was evaluated by a comparison of q-PCR data with manual counts of
*Phaeocystsi pouchetii* in field samples (data not shown). In the first step total eukaryotic 18S rDNA was
amplified from a positive control (genomic DNA *Phaeocystis pouchetii*), a negative control (no template) and
genomic DNA isolated from field samples using the universal primer-set 1F-(5′-AAC TGG TTG ATC CTG
CCA GT-3′) / 1528R- (5′-TGA TCC TTC TGC AGG TTC ACC TAC-3′) (modified after Medlin et al., 1988).
PCR-amplifications were performed in a 20 µl volume in a thermal cycler (Eppendorf, Germany) using 1x
HotMasterTaq buffer containing $Mg^{2+}$, 2.5 mM (5′Prime); 0.5 U HotMaster Taq polymerase (5′Prime,
Germany); 0.4 mg/ml BSA; 0.8 mM (each) dNTP (Eppendorf, Germany); 0.2 µM of each primer (10 pmol/µl)
and 1µl of template DNA (20 ng/µl). The amplification was based on 35 cycles, consisting of 94°C for 1 min,
54°C for 2 min and 72°C for 2 min, followed by 1 min denaturation at 94°C and finalized by a final extension of
10 min at 72°C. Subsequently PCR products were purified using the QIAquick PCR purification kit (Qiagen,
Hilden, Germany).  In the second step a qPCR-assay was carried out using a species specific primer-set 82F-(5′-
GTG AAA CTG CGA ATG GCT CAT-3′) / P1np- (5′-CGG GCG GAC CCG AGA TGG TT-3′) for
*Phaeocystis pouchetii*. The quantitative PCR-assays were performed in triplicate in a 20 µl volume in a 7500
Fast Real-Time PCR-System (Life Technologies Corporation; Applied Biosystems, USA) using 1x SYBR Select
Mastermix (Life Technologies, USA); 0.2 µM of each primer (10 pmol/µl) and 2µl of the purified 18S rDNA
PCR-fragment. The amplification was based on 40 cycles, consisting of 95°C for 10 min, 95°C for 15 sec, 66°C
for 1 min. The quantitative PCR-assay was calibrated with a dilution series of a laboratory culture of *Phaeocystis*
*pouchetii* (Figure 4). Based on this calibration CT-values were transformed into cell numbers using the following
equation: CT= -2.123 ln (cell numbers) + 38.788.

**3 Results and Discussion:**
**3.1 Overview Molecular Based Observation Strategy**
The molecular based observation strategy that we present here is organized in 4 different levels (Figure 1). At
level 1, samples are collected in high spatio-temporal resolution using the remote-controlled automated filtration
system AUTOFIM (Figure 2). The sampling system can either be deployed on a fixed monitoring platform or
aboard a ship (research vessel or ship of opportunity) without the need of highly trained personal. Samples can
be preserved with a preservation buffer (e.g. DNAgard, Biomatrica, USA) for later laboratory analyses, or
directly subjected to molecular surveillance of key species aboard the ship via an automated biosensor system or
quantitative polymerase chain reaction (level 2). Direct analyses aboard ships provide near real time information
on abundance and distribution of phytoplankton key species that can be used to optimize phytoplankton
sampling for detailed high resolution analyses of overall phytoplankton composition during an ongoing sampling
campaign. The resulting preserved samples will be analyzed at the next observational levels in the laboratory
(level 3 and 4). This involves at level 3 molecular fingerprinting methods that provide a quick and reliable
overview of differences in protist community composition of the samples in a given observation area or time
period. Furthermore, this information can be used to select representative samples for detailed analysis of
taxonomic protist composition via latest next generation sequencing at level 4. An overall integrated dataset of
the results based on the different analyses provides comprehensive information on the diversity and
biogeography of protists, including all related size classes. At the same time, the cost effort of the observation is
optimized in respect to analysis effort and time. Sampling based on the autonomous filtration device is more cost
efficient, because labor costs and the requirement of ship space and time are reduced.
The development of the Molecular Observation Strategy was based on extensive method development
and evaluation. Overall, it included: (i) the development of an automated remote controlled filtration system
(Figure 2), (ii) the evaluation and application of **A**utomated **R**ibosomal **I**ntergenic **S**pacer **A**nalysis (ARISA)
(Kilias et al., 2015), (iii) the implementation of Next Generation Sequencing (454-pyrosequencing; Illumina) for
marine protists (e.g. Wolf et al., 2013) and (iv) the development and evaluation of molecular probe based
methods such as molecular sensors (Wollschlaeger et al., 2014) or quantitative PCR (qPCR). Most of the field
work presented here in this publication was accomplished in the Arctic Ocean with special emphasis on the area
of the "Deep-Sea Long-Term Observatory Hausgarten" established by the Alfred Wegener Institute for Polar-
and Marine Research in 1999 to carry out regular observations of the ecosystem in the eastern Fram Strait
(Soltwedel, 2005). In the following, the different parts of the observation strategy are presented in detail.
**3.1.1 Automated remote controlled filtration system**
The remote controlled **auto**mated **fil**tration system for **m**arine microbes (AUTOFIM) is the core of the
observation strategy. The filtration system (Figure 2) can be operated autonomously aboard research vessels or
ships of opportunity. AUTOFIM allows filtration of a sampling volume up to five litres from the upper water

column. In total, 12 filters can be taken and stored in a sealed sample archive. Prior to storage, a preservative such as Lysis Buffer RLT (Qiagen, Germany) is applied to the filters preventing degradation of the sample material, that can be used for molecular or biochemical analyses. Exchanging the sample archive is a quick and easy task, which makes it feasible for lay persons from the ships´ staff to take care of the automated filtration. This would circumvent the need to provide support of an additional specifically trained personal for filtration in the field. Filtration can be triggered after defined regular time intervals or remote controlled from a scientist at the research institute. Additionally, it could also be event-triggered if the filtration system would be operated in connection with *in situ* sensor systems (Petersen, 2014). Overall, AUTOFIM provides the technical background for automated high spatio-temporal resolution collection of marine particles e.g. for molecular analyses. During expedition PS92 of RV Polarstern to the Arctic Ocean in summer 2015, AUTOFIM was used for the first time to collect samples from the upper water column at a depth of ~ 10 m, which is the depth of the inlet of the ships water pump system. Subsequent to filtration, samples were preserved with a preservation buffer and stored at -80°C until further analyses in the laboratory.

**3.1.2 Automated Ribosomal Intergenic Spacer Analysis (ARISA)**

ARISA provides information on variability in protist community structure in larger sample sets at reasonable costs and effort. In an ARISA-analysis the community is characterized by its community profile, which is based on the composition (presence/absence) of differently sized DNA fragments. The DNA fragments are a result of the amplification of the internal transcribed spacer region of the ribosomal operon, which displays a high degree of taxon-related variability in its length. ARISA-profiles reflect taxon specific differences observed in NGS-data sets (Kilias et al., 2015). In the developmental phase of the molecular observation strategy, this method was used in a number of different studies to better understand variability of Arctic marine protist communities in relation to environmental conditions and ocean currents. Based on ARISA analyses we identified large scale patterns of protist biogeography that were tightly connected to ambient water masses, ocean currents and sea ice coverage (Kilias et al., 2014a; Metfies et al., 2016; Wolf et al., 2014b). We suggest using ARISA as part of the molecular observation strategy to identify biogeographic or biodiversity patterns in large sample sets, e.g. collected via AUTOFIM. Identification of pattern in phytoplankton biogeography or biodiversity requires analyses of large samples sets, because spatial heterogeneity of marine phytoplankton is considerable, while the vertical dimension is of particular importance, since differences in vertical abundance and composition of phytoplankton impact primary production, export processes and energy transfer to higher trophic levels (Leibold, 1990). Vertical distribution of marine protists is determined by opposing resource gradients and mixing conditions (Mellard et al., 2011). In respect to this it was necessary to evaluate how representative samples from 10m depth might be for the photic zone in the underlying water column. This would be important in case AUTOFIM would be applied to study large scale biogeographic patterns of marine protists. Acknowledging the potential of ARISA to quickly generate meaningful information on variability between protist samples, we used this methodology in this study to assess the similarity of phytoplankton community composition in samples from the upper water column collected with AUTOFIM and in samples collected in deeper water layers via CTD at the same location. The ARISA patterns obtained from deeper water layers (20m; 50m) are highly similar to those obtained from the samples collected with AUTOFIM. The samples collected with AUTOFIM at stations PS92/19 and PS92/43 clustered together with the individual samples collected at other depths at the same

location (5m; 20m; 50m) and with the integrated signal from the CTD sampling (all three depths) at this location
(Figure 3). This result suggests that qualitative information on phytoplankton community composition based on
sampling with AUTOFIM can be considered as being representative for the photic layer of the water column.
This might be attributed to the observation that geography and ambient water masses have a major impact on
qualitative composition of marine plankton communities on a larger scale, with plankton communities being
partially structured according to the basin of origin (de Vargas et al., 2015; Metfies et al., 2016).
**3.1.3 Next Generation Sequencing (454-pyrosequencing; Illumina)**
Sequencing of ribosomal genes is a valuable approach to describe the taxonomic composition of protist
communities including the small size fractions. Technical progress in this field has been tremendously rapid over
the last 5-10 years. Around ten to fifteen years ago, sequencing of 18S rDNA clone libraries was the gold
standard to assess marine eukaryotic and prokaryotic communities (Hugenholtz, 2002). Around six years ago,
first studies reported the use of 454 pyrosequencing for assessment of prokaryotic diversity (Turnbaugh et al.,
2009). The massively parallel 454-pyrosequencing was found to generate several hundred thousands of
ribosomal sequence per sample and had the potential to uncover more organisms, even rare species from large
scale biodiversity surveys (Sunagawa et al., 2015). We assessed the validity of 454-pyrosequencing by
evaluating the sequence data sets with results obtained via other methods, such as 18S clone libraries, HPLC and
microscopic counts. The samples analyzed in the course of this evaluation originated from the same Niskin-
bottle of a respective CTD-cast. In our data sets pyrosequencing data were in good agreement with information
on community composition generated by high pressure liquid chromatography (HPLC) or clone libraries (Kilias
et al., 2013; Wolf et al., 2013). During the past six years, we used 454-pyrosequencing to determine the
variability of protist community structure in Fram Strait, in the area of the "Deep-Sea Long-Term Observatory
Hausgarten", and the central Arctic Ocean (Kilias et al., 2014a; Metfies et al., 2016). Overall, our data revealed
that *Phaeocystis pouchetii* is an important contributor to Arctic protist communities, particularly to the pico-
eukaryote community composition. In 2009 the species constituted up to 29.6% of the sequence assemblage
retrieved from pico-eukaryote samples in that area (Kilias et al., 2014b). A larger survey of Arctic protist
community composition in 2012 including Fram Strait and larger parts of the Central Arctic Ocean confirmed
these observations and identified *Phaecystis pouchetii* once again as an important contributor to Arctic pico-
eukaryote Chl *a* biomass. The latter constituted between 60-90% of Chl *a* biomass during summer 2012 in the
Arctic Ocean (Metfies et al., 2016). This comprehensive sequence based information on phytoplankton
community composition was very well suited to serve as a basis for the development of molecular probes that
can be used for molecular surveillance with molecular sensors or quantitative PCR (qPCR).
**3.1.4 Development and evaluation of molecular probe based methods: molecular sensors or qPCR**
Molecular Sensors are chip-based formats that allow parallel identification and quantification of multiple taxa in
a single experiment. The identification is based on solid phase hybridization of molecular probes, immobilized to
the surface of the sensor chips that bind to the rRNA or rDNA of the target species (Diercks et al., 2008a;
Diercks et al., 2008b; Ussler et al., 2013). Quantitative or real time PCR (qPCR) is a PCR-based method that
utilizes fluorescent dyes or fluorescently-labelled molecular probes to quantify nucleic after each PCR cycle. It is
a useful tool for quantitation of nucleic acids, respectively species in a given environment (Toebe et al., 2013).

An automated molecular sensor (Diercks et al., 2008a) and qPCR are intended to be part of the molecular observation strategy in order to generate near real time information on the occurrence of key species on board ship and to complement NGS-based information on phytoplankton community composition with quantitative information on the occurrence of selected key species (Figure 1). These approaches are necessary because of biases related to the amplification of the 18S rDNA gene via PCR, and uncertainties in respect to copy number of the gene in the genome of different species (e.g. Zhu et al.,2005), which make it difficult to deduce species abundance based on NGS. We developed new molecular probes for relevant taxa that were major contributors in our NGS-libraries or that were known from published literature to occur in the observation areas (North Sea and Arctic Ocean). The molecular probes were either used in combination with molecular sensors (Wollschlaeger et al., 2015), qPCR or fluorescent *in situ* hybridization (FISH) (Thiele et al., 2014). The data on species abundance obtained from  of molecular sensors targeting either 18S rDNA or 18S rRNA were evaluated with the results obtained from microscopic counts (Wollschlaeger et al., 2014). The molecular sensor targeting 18S rRNA shows a robust linear relationship between molecular sensing signal and cell counts via microscopy. The positive evaluation results for the rRNA based nucleic acid biosensor suggest an excellent potential of the method to be used as module in a Molecular Observation Strategy. Here, the regular quantitative molecular monitoring would benefit from advantages like reduced effort (time, costs and labour), and the high potential for automation of the methodology (Wollschlaeger et al., 2014). In this study we demonstrate the potential of quantitative PCR to better understand the biogeography and abundance of *Phaeocystis pouchetii* in Arctic Waters using a specific primer set for qPCR. The qPCR values were calibrated against defined numbers of laboratory cultures (Figure 4) to allow quantification of *Phaeocystis pouchetii* via this method. During expedition PS85 of RV Polarstern in June 2014, we used qPCR on board ship to determine the abundance of *Phaeocystis pouchetii* on a transect through Fram Strait at ~79°N (Figure 4). The results of our survey suggest that abundance of *Phaeocystis pouchetii* in Fram Strait is determined by water mass properties such as salinity, ice coverage and water temperature. Salinity is positively correlated with abundance of *Phaeocystis pouchetii*. The abundance of *Phaeocystis pouchetii* was higher in Atlantic Waters, which are characterized by higher salinities in the range of 33-34 PSU than in Polar Waters of Fram Strait which are characterized by salinities around 31 PSU. In Atlantic Waters the average cell number of *Phaeocystis pouchetii* was ~ 3.5 times higher than the average cell number in Polar Waters of Fram Strait. Furthermore, Chl *a* biomass appears to be correlated with abundance of *Phaeocystis pouchetii*. Our findings are in agreement with previous studies that reported blooms of *Phaeocystis pouchetii* in waters around Svalbard with cell abundances in a similar range as observed in this study (Wassmann et al., 2005). In 2012, we carried out a large scale study to survey the biogeography of marine protists in the Arctic. This survey included a comprehensive NGS based analysis of community composition along 79°N in Fram Strait in June and later in the season in Nansen Basin and Amundsen Basin. Overall, the findings of 2014, suggesting a positive correlation of Atlantic water properties, e.g. higher salinity and lower ice coverage with high abundance of *Phaeocystis pouchetii* are in agreement with the previous study of 2012. This study also found a positive correlation in agreement with the findings of 2014, even though sequence abundance of *Phaeocystis pouchetii* was more evenly distributed in Fram Strait in 2012 (Metfies et al., 2016). This might be attributed to the complex current system in the area. Overall, qPCR carried out on board ship provided a near real time overview of the distribution of a protist key species during expedition PS85.

## 4 Conclusions

Here we introduce for the first time an integrated hierarchically organized molecular based observation strategy that combines autonomous sampling with molecular analyses. It is a valuable tool to survey phytoplankton abundance and biodiversity in the desired high spatial and temporal resolution as well as at different levels of taxonomic resolution. The observation strategy is based on a combination of ship based automated filtration, online measurements of oceanographic parameter, and different molecular analyses. On one hand, our approach provides near real time information on phytoplankton key species abundance in relation to environmental conditions already on board ship. On the other hand, it provides detailed information on variability in the total phytoplankton community composition based on comprehensive, laboratory-based molecular analyses such as molecular fingerprinting methods and NGS. This information can be subsequently correlated with information on the physical and chemical marine environment and has excellent potential to complement other hierarchically organized observation strategies as described e.g. for the detection of marine hazardous substances and organisms (Zielinski et al., 2009). In summary, our molecular observation strategy is a significant contribution to refine regular assessment of consequences of ongoing environmental change for marine phytoplankton communities with respect to adequate spatial, temporal, and taxonomic resolution.

## 5 Acknowledgements

This work was supported through the Coastal Observing System for Northern and Arctic Seas (COSYNA), by institutional funds of the Alfred Wegener Institute for Polar- and Marine Research, Bremerhaven, funds of the Helmholtz Zentrum Geesthacht Centre for Materials and Coastal Research, and funds of the Initiative and Networking Fund of the Helmholtz Association for financing the Helmholtz-University Young Investigators Group PLANKTOSENS (VH-NG-500). We thank the crew of R.V. *Polarstern* for excellent support during the work at sea. Furthermore we thank Kerstin Oetjen, Swantje Rogge and Christiane Lorenzen for great technical assistance. Annegret Müller and Uwe John are acknowledged for excellent technical support of the fragment analysis.

**Figure Legends:**

Fig. 1: A:Overview of the smart observation strategy which is organized in four different levels: level 1: samples are collected underway or at monitoring sites using the remote-controlled automated filtration system AUTOFIM; level 2: direct molecular surveillance of key species aboard the ship via an automated biosensor system or quantitative polymerase chain reaction; level 3:.preserved samples are analyzed via molecular fingerprinting methods (e.g. ARISA) that provide a quick and reliable overview of differences in protist community composition of the samples in a given observation area or time period; level 2: detailed analysis of taxonomic protist composition in selected samples via latest next generation sequencing. B-E: Schematic diagrams illustrating the analyses used in the smart observation strategy.

Fig. 2: A: AUTOFIM installed on board RV Polarstern (1: Sample reservoir; 2: Filtration; 3: Archive for preserved filters. B: Filtration-module (1:Filter stacker; 2:Filtration cap).

Fig. 3: MetaMDS Plot (non metric multidimensional scaling plot) of ARISA fingerprints generated from samples collected via Niskin bottles coupled to a CTD-rosette and AUTOFIM.The closer the samples are located to each other in the metaMDS-plot, the more similar are the ARISA-profiles of the samples. The label of the samples gives information on the cruise leg (PSXX) and the station (/XX). Samples were collected during expeditions PS92 and PS 94 of RV Polarstern to the Arctic Ocean during summer 2015. The samples collected during PS94 serve as an outgroup in this analysis.

Fig. 4: Assessment of *Phaeocystis pouchetii* in Fram Strait. A: Calibration of *Phaeocystis pouchetii* specific qPCR assay with a dilution series of laboratory cultures. The CT value is significantly correlated with cell numbers. B: Abundance of *Phaeocystis pouchetii* in Fram Strait. The dots and the associated numbers represent sampling sites and associated station numbers of expedition ARKXXVIII(PS85) of RV Polarstern in summer 2014, while cell numbers/liter are reflected by different colours. C: Principal component analysis including environmental parameters (temperature, salinity, Chl *a* biomass and sea ice coverage) and abundance of *Phaeocystis pouchetii*. Triangles and associated numbers represent sampling sites and associated station numbers of expedition ARKXXVIII (PS85) of RV Polarstern in summer 2014. HG4 indicates the central station of the "Deep-Sea Long Term Observatory Hausgarten" in Fram Strait. The Eigenvalues indicate the proportion of variance explained by different dimensions in the diagram. The black bars in the histogram reflect the x-axis and the y-axis. Here ~ 80% of variance is explained in this two-dimensional diagram of the PCA (x-axis: 50.29%; y-axis: 30.08%).

**Figure 1:**

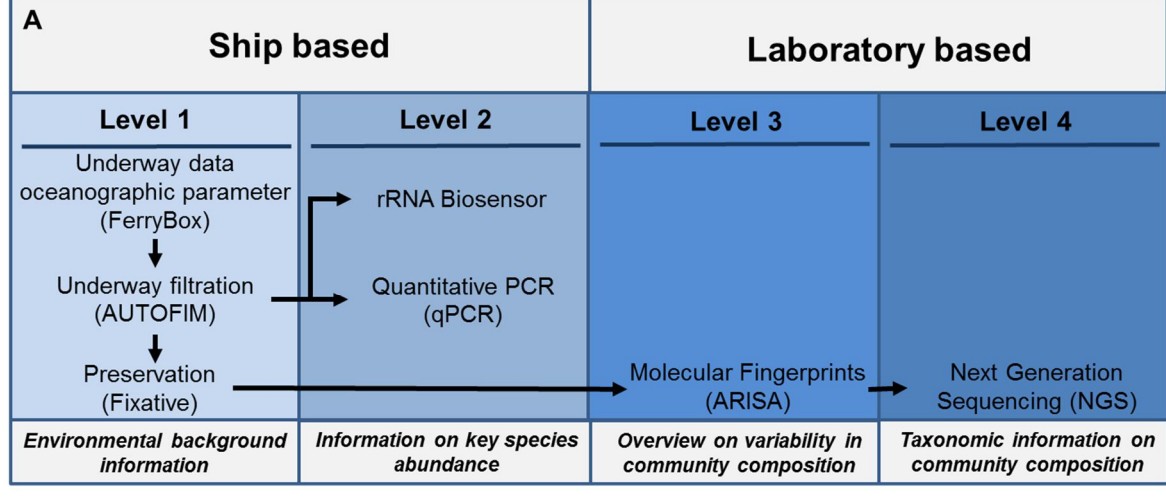

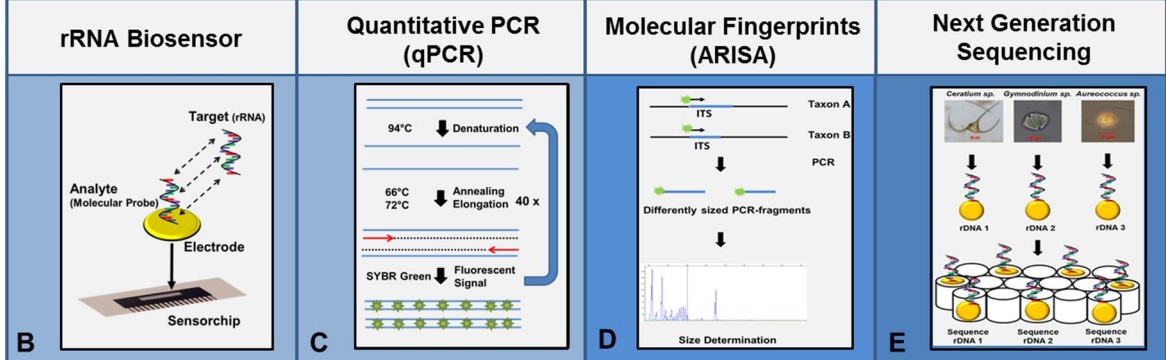

**Fig. 1:** Fig. 1: Overview of the smart observation strategy which is organized in four different levels: level 1: samples are collected underway or at monitoring sites using the remote-controlled automated filtration system AUTOFIM; level 2: direct molecular surveillance of key species aboard the ship via an automated biosensor system or quantitative polymerase chain reaction; level 3:.preserved samples are analyzed via molecular fingerprinting methods (e.g. ARISA) that provide a quick and reliable overview of differences in protist community composition of the samples in a given observation area or time period; level 2: detailed analysis of taxonomic protist composition in selected samples via latest next generation sequencing. B-E: Schematic diagrams illustrating the analyses used in the smart observation strategy.

**Figure 2:**

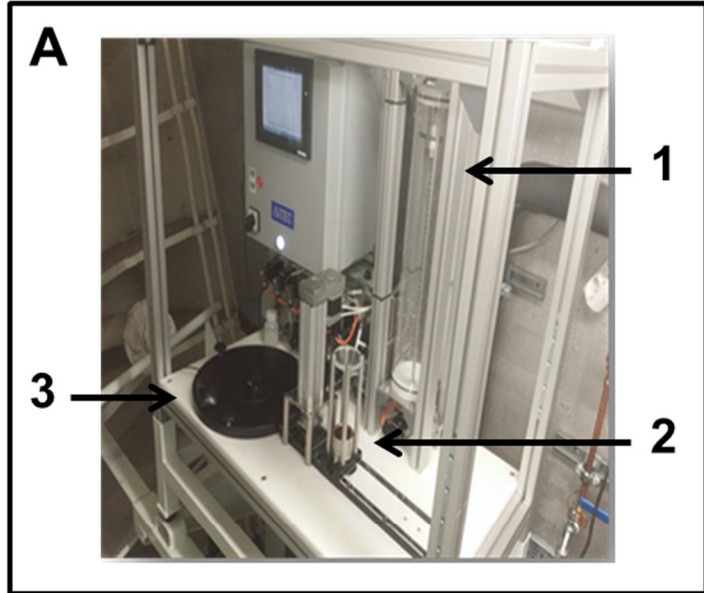

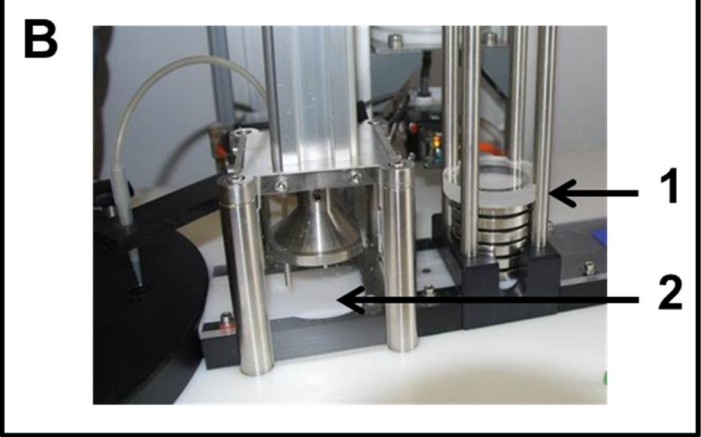


**Fig. 2:** A: AUTOFIM installed on board RV Polarstern (1: Sample reservoir; 2: Filtration-module; 3: Archive for
preserved filters. B: Filtration-module (1:Filter stacker; 2:Filtration cap).
















**Figure 3:**

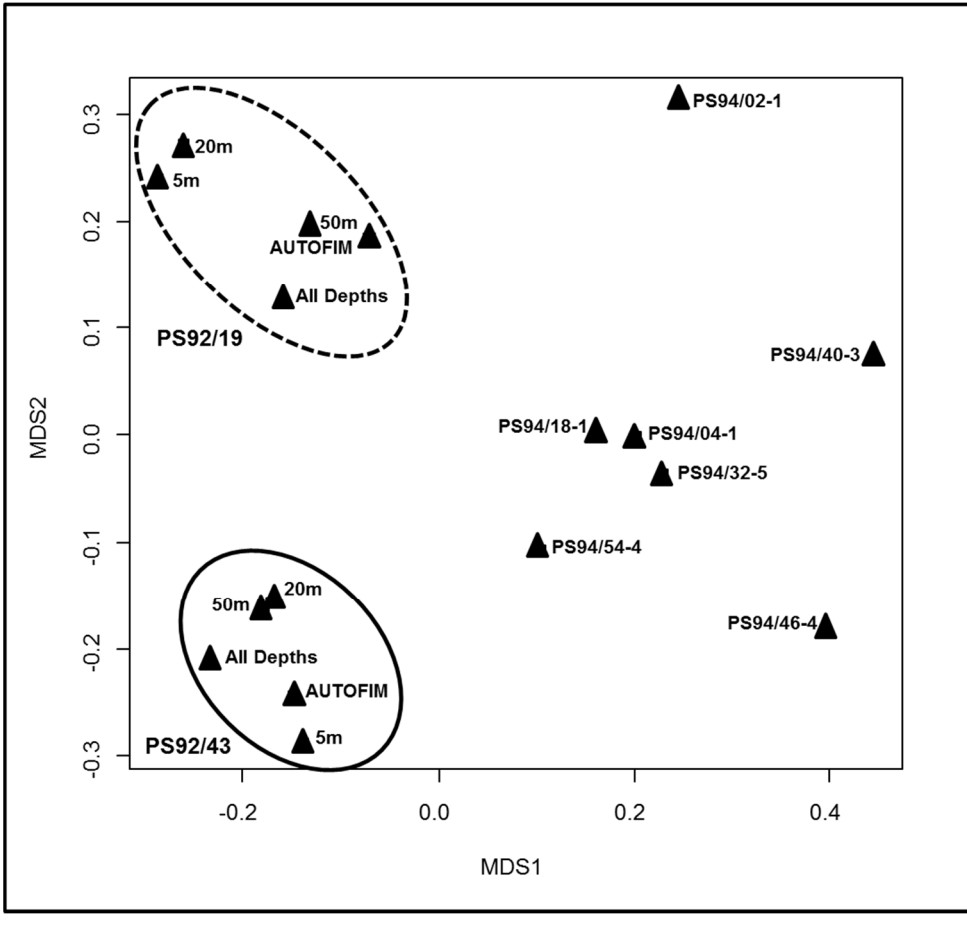

**Fig. 3:** MetaMDS Plot (non metric multidimensional scaling plot) of ARISA fingerprints generated from samples collected via CTD and AUTOFIM.The closer the samples are located to each other in the metaMDS-plot, the more similar are the ARISA-profiles of the samples.The label of the samples gives information on the cruise leg (PSXX) and the station (/XX). Samples were collected during expeditions PS92 and PS 94 of RV Polarstern to the Arctic Ocean during summer 2015. The samples collected during PS94 serve as an outgroup in this analysis.

**Figure 4:**

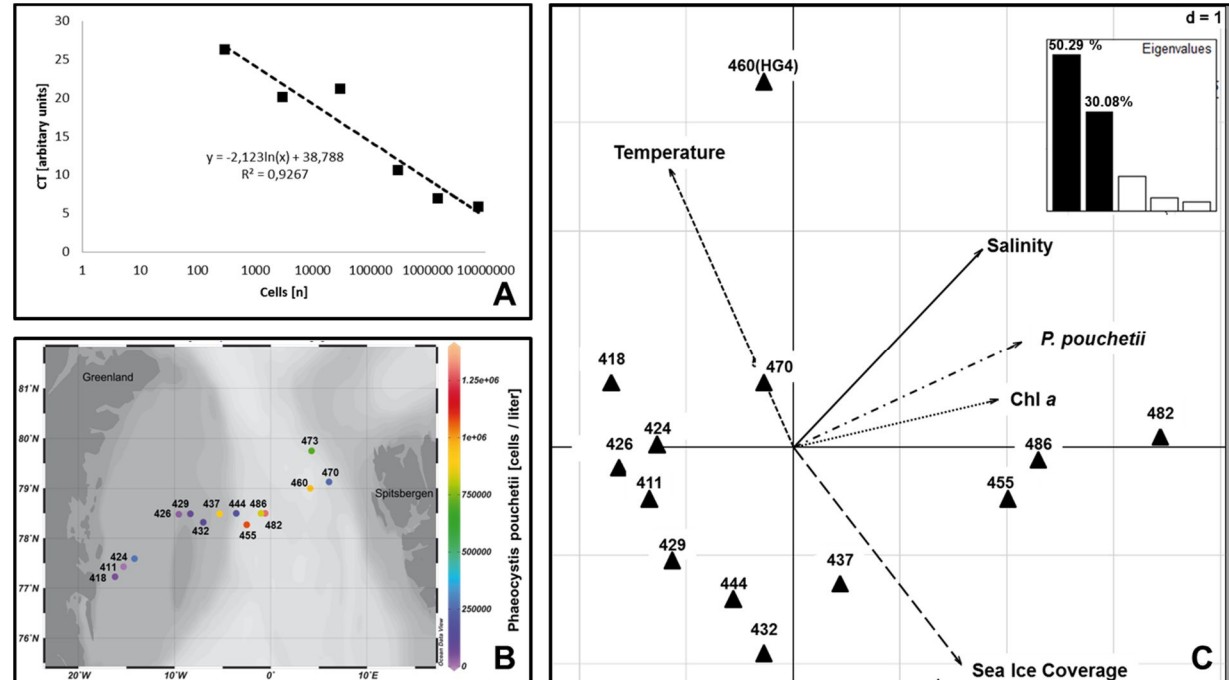

**Fig. 4:** Assessment of *Phaeocystis pouchetii* in Fram Strait. A: Calibration of *Phaeocystis pouchetii* specific qPCR assay with a dilution series of laboratory cultures. The CT value is significantly correlated with cell numbers. B: Abundance of *Phaeocystis pouchetii* in Fram Strait. The dots and the associated numbers represent sampling sites and associated station numbers of expedition ARKXXVIII(PS85) of RV Polarstern in summer 2014, while cell numbers/liter are reflected by different colours. C: Principal component analysis including environmental parameters (temperature, salinity, Chl *a* biomass and sea ice coverage) and abundance of *Phaeocystis pouchetii*. Triangles and associated numbers represent sampling sites and associated station numbers of expedition ARKXXVIII (PS85) of RV Polarstern in summer 2014. HG4 indicates the central station of the "Deep-Sea Long Term Observatory Hausgarten" in Fram Strait. The Eigenvalues indicate the proportion of variance explained by different dimensions in the diagram. The black bars in the histogram reflect the x-axis and the y-axis. Here ~ 80% of variance is explained in this two-dimensional diagram of the PCA (x-axis: 50.29%; y-axis: 30.08%).

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
