# Peer review of "High resolution monitoring of marine protists based on an observation strategy integrating automated on board ship filtration and molecular analyses Katja Metfies1, Friedhelm Schroeder2, Johanna Hessel1, Jochen Wollschläger2, Sebastian Micheller1, Christian Wolf1, Estelle Kilias1, Pim Sprong1, Stefan Neuhaus3; Stephan Frickenhaus3 Wilhelm Petersen2 1Helmholtz Young Investigators Group PLANKTOSENS,"

_Ocean Science, 2016_

## Referee Comment (RC1) · Anonymous Referee #1 · 22 Jun 2016

L30 The observation strategy is organized AT four different levels. L30 At level 1, samples are collected AT.... L35 protist mentioned for first time here and the first sentence relates only to photosynthetic microbes. Ensure it is defined earlier or use another consistent term. L36 replace subjected with used L36-37: via THE latest next-generation sequencing TECHNOLOGY... L63: Microplankton should have an upper limit. L66 Unclear what "currently driving topics" means. Do you mean its a topical issue in marine ecology? L74 reference difficulties in assessing pico/nano sized fraction needed. L87-88: I would say currently restricted to mostly monitoring larger phytoplankton (it does record coccolithophores that are nano-sized). L105 suggest an alternative starter: TO

ADDRESS THESE ... L108 remove "a large variety of", for THE observation L121 end the sentence with something like "to identify protists". L126 define NGS L154: Here, two LITRES OF water. Does the fitting of AUTOFIM require expertise? L153-157: Which parts of this process is carried out manually or required scientific supervision and which automatically by AUTOFIMS? How is it cleaned (if at all?) between samples? L155: Was this done by AUTOFIM L180-184: More information needed on bioinformatic methods or reference to methods. Methods for any specific comparisons in the results need to be elaborated, e.g. "our data sets pyrosequencing data were in good agreement with information on community composition generated by high pressure liquid chromatography (HPLC) or clone libraries..". Explain here how HPLC comparison L184 remove e.g if these articles have sufficient information L186: Why was a nested approach used, this may have implications for the quantification step- how was this overcome? L195-202: What controls did you use? L203: where does this equation come from? L235-245: So can it be deployed without any experts on it from start to finish? L238- Name the preservative L248- How did you ensure the piping in the ship pump apparatus was clear of microbial biofilm and/or residual water? L249 "...at meso- or large in large sample sets". Sentence confusing- meso or large-scale- you mean geographically large or large sample numbers? Re-word if its due to large sample numbers explain why? L260 alter to....marine phytoplankton is CONSIDERABLE L260-1 "...dimension is OF particular importance," L258-263: The word However is used but i cannot see a connection between the first sentence and the second. Clarification needed. L270: What do you mean by "deeper horizons"? do you mean greater depths? I think you need to explain the connection between autofim stations and ctd stations-where they geographically close or was it just depth? I would mention that 5m- 50m is within the photic zone. Also in what way were they similar- taxonomic assemblage or together with other factors? L271 typo in individual L272 ..and WITH the integrated signal FROM THE CTD SAMPLING at all three depths.... L277: According to THE basin of origin L282- 284: This is extra information so can go. L284: "...parallel 454-pyrosequencing WAS FOUND TO generate..." L286-7: What sequence data sets?

CTD, AUTOFIM or both? L290: to determine THE variability L295: Repetitive. Replace "collected in the area of the "Deep-Sea Long-Term Observatory Hausgarten"" with "in that area". L304: Alter to "Development and evaluation of molecular probe based methods: molecular sensors and qPCR" L307: "...the surface of the sensor chips THAT BIND to EITHER the rRNA (transcriptome) or rDNA (genome) of the target species." L306:308: Mention that it is also quantitative and how this is achieved- a diagram of these methods would help readers understand. L308-310: Alter to "Quantitative or real time PCR (qPCR) IS A PCR-BASED METHOD THAT UTLITISES FLUORESCENT DYES OR FLUORESCENTLY-TAGGED DNA PROBES TO QUANTIFY SPECIES BY DETECTING THE AMOUNT OF DNA FORMED AFTER EACH PCR CYCLE". This way the reader can link species abundance with DNA quantity. L310: Also useful for quantifying species. L314-317: reference needed for this sentence L317: May be use "As such" or another term instead of "In respect to this," L322: What are you measuring "from microscopy, HPLC and flow cytometry". Cell counts, pigments? Add this in. How did you related pigments to cell counts- give a reference to the method? L323: What about the other measurements? L324 replace high potential with good potential L325: What do you mean by "the related regular monitoring". Is it qpcr/molecular sensors? If so i suggest Here, additional quantitative molecular monitoring L326: reduced effort in what? Change high potential to excellent or good potential L333: delete while L335: PSU, than.... Delete comma. L343 reference needed for the 2014 findings L345: This study also suggested this positive correlation. Suggest This study also found a positive correlation in agreement with XYZ, et al 2014. L352: hierarchically organized molecular based. I would add that its a combined autonomous sampling and molecular testing platform. L360 change strong to excellent/good

Figures I think map figure would be really helpful to allow readers to understand spatial comparisons Also for section 3.1.4 for readers who are not familiar with molecular methods a diagram of how a qpcr/molecular sensors work or a photograph of the one you have would be good- you could alter fig 1 as its quite small and provide a clearer picture of these? Fig. 2: Would be good to see basic diagram of its layout and its
connected and its modules. Fig. 3: Define Meta MDS. Methodology needs to be referred to in methods. I would explain the labelling system for expeditions and stations. Where/when were the other PS stations and what did they represent? Fig. 4: A- i would explain the significance of that graph to non-expert readers, to say the assay worked and provided a good relationship between DNA quantity and cell abundance. B- What do the numbers next to the points in the map represent? C: parameter needs to be plural,. What does the inset graph show? What do the numbers represent by the triangles? I suggest explain the plot.

---

## Referee Comment (RC2) · Anonymous Referee #2 · 8 Aug 2016

The manuscript entitled "High resolution monitoring of marine protists based on an observation strategy integrating automated on board ship filtration and molecular analyses" by Katja Metfies and colleagues introduce the possibility of high resolution, automated sampling of seawater that can later be analysed to assess microbial diversity and abundance based on molecular tools. The automated sampling and filtration equipment that has been developed and tested allow for new sampling possibilities from ships to assess e.g. biological responses to environmental variability.

Comments: In general the manuscript is well-written and clear, discussing the possibilities of the new methodology. In particular the AUTOFIM system provides a highly relevant sampling possibility for research vessels allowing automated sampling and conservation of filtered organic material for molecular analyses. Some parts of the methodology have been tested previously, and the current manuscript gives a nice compilation of those studies. A bit more thorough overview of related technology could be relevant, however. For instance the ESP (Environmental Sample Processor) has an automated filtration unit directly connected to the possibility of using qPCR or hybridisation for microbial species identification under water (although not as part of a ferry box system as far as I know).

Regarding the test performed to assess how representative the 10m sample taken by AUTIFIM is for the underlying water column: Firstly, the authors should explain what they mean by "underlying water column". They did not present vertical CTD profiles of the water column, so it is difficult to know if the samples taken from distinct depths using the Niskin bottles were all taken from the same layer. Assuming that the samples collected for this test was from the same layer: Their results show that at the (only) two stations sampled for the comparison, the AUTOFIM sample communities at 10m were associated with the communities collected using Niskin bottles from the same water layer. Their discussion around this result (around lines 270-274) is a bit unclear referring how the AUTOFIM samples is representative of that of "deeper horizons". What do the authors mean by this? I assume they mean that the AUTOFIM samples are representative for the upper mixed layer, because they do not have data from deeper layers. This part should be rephrased/clearified.

The automated biosensor system used - it is a bit unclear to me exactly what this is. Do they refer to ferry box data or to analyses performed on the filtered seawater collected using the AUTOFIM? If the latter, which sensors/analyses do they refer to? Or do the refer to the molecular sensors of Wollschlaeger et al. (2014)?

In the introduction, when the authors refer to the molecular tools available (line 105 and onwards), they mainly refer to their own work. But there are several studies that would

be relevant to include in such an overview. I suggest the authors refer also to other studies from Arctic waters, in particular the Canadian Arctic has been explored using similar and relevant molecular tools.

Typos and minor comments: lines 97-99: This statement should include references.

line 152: Were the particles for molecular analyses added a buffer? Or perhaps stored in -80 prior to DNA extraction?

line 165: Is the use of the E.Z.N.A DNA extraction kit correct? And was it used for both the AUTOFIM and Niskin bottle samples? If so, why was the the Qiagen lysis buffer added to the filters collected using AUTOFIM?

line 178: ITS1 is the internal transcribed spacer 1. It is also an "intergenic spacer region", but the use of that term without explaining the ITS1 abbreviation is a bit confusing.

The first paragraph of 3.1 is mostly repeating what is already pointed out in the introduction. This section could be reduced.

line 259: Rephrase sentence, the word "scale" lacking? line 261: "from" should be replaced with "of" (particular importance)

line 297-299, incl Metfies et al 2016: Is the % cells of Phaeocystis due to % reads or quantitative counts? If it refers to % reads, the statement is a bit strong.

Fig. 1 text: This text should explain the different levels of the observation strategy in greater detail, so that it is not necessary to check the manuscript text to identify the different parts.

Fig. 3 text: Samples collected via CTD ... imprecise, the samples were collected using Niskin bottles.

Fig. 4 text: This text also does not explain the figure very well. Do the numbers represent station numbers? The eigenvalues histogram in 4C is not explained - what

do the black vs white histograms signify? What values are at the y axis?

---

## Author Comment (AC1) · 30 Aug 2016

Author's Reply (AR) to Anonymous Referee # 1 (R1) R1: L30 The observation strategy is organized AT four different levels AR: The sentence was changed according to the reviewer's suggestion.

R1: L30 At level 1, samples are collected AT... AR: The sentence was changed according to the reviewer's suggestion.

R1: L35 protist mentioned for first time here and the first sentence relates only to photosynthetic microbes. AR: The term protist is introduced now in the first sentence

of the abstract (L 24 Information on recent biomass distribution and biogeography of photosynthetic marine protists)

R1: L36 replace subjected with used AR: The sentence was changed according to the reviewer's suggestion.

R1: L36-37 via THE latest next-generation sequencing TECHNOLOGY AR: The sentence was changed according to the reviewer's suggestion.

R1: L63 Microplankton should have an upper limit AR: An upper limit was added (L63(20-200 $\mu$m))

R1: L66 Unclear what "currently driving topics" means. Do you mean its a topical issue in marine ecology? AR: L66 The term "driving topics" was replaced by "topical issues"

R1: L74 reference difficulties in assessing pico/nano sized fraction needed. AR: An exemplary reference was added (Caron et al., 1999).

R1: L87- 88: I would say currently restricted to mostly monitoring larger phytoplankton (it does record coccolithophores that are nano-sized). AR: The sentence was changed according to the reviewer's suggestion (L88-89 Unfortunately, the CPR-approach is restricted to zooplankton and larger phytoplankton e.g. diatoms. Again, the ecological relevant picophytoplankton fraction is omitted).

R1: L105 suggest an alternative starter: TOADDRESS THESE ... AR: The sentence was changed according to the reviewer's suggestion (L107 To address these shortcomings and challenges of current observation approaches....) R1: L108 remove "a large variety of", for THE observation AR: The sentence was changed according to the reviewer's suggestion (L109-110 Over the past decade numerous publications demonstrated the power of molecular methods for the observation of marine plankton organisms).

R1: L121 end the sentence with something like "to identify protists". AR: The sentence was changed according to the reviewer's suggestion.

R1: L126 define NGS AR: The sentence was changed according to the reviewer's suggestion (L128 . In contrast, Next Generation Sequencing (NGS) of ribosomal genes)

R1:L154: Here, two LITRES OF water. AR: The sentence was changed according to the reviewer's suggestion (L152 Two liter of water subsamples were taken in PVC bottles. . .)

R1: L153-157 - Does the fitting of AUTOFIM require expertise? - Which parts of this process is carried out manually or required scientific supervision and which automatically by AUTOFIMS? -Was this done by AUTOFIM ? AR: A sentence to clarify these questions was added (L157-160 Fitting and programming of the device does not require special expertise if it is done according to the manufacturer's protocol. All steps related to the filtration process, including application of Lysis Buffer RLT (Qiagen, Germany) were carried out automatically by AUTOFIM.)

R1: L153-157 How is it cleanded (if at all?) between samples? AR: A sentence to clarify this question was added (L162-163 The filtration device was cleaned after each filtration step by rinsing the device with fresh-water.)

R1: L180-184: More information needed on bioinformatic methods or reference to methods. Methods for any specific comparisons in the results need to be elaborated, e.g. "our data sets pyrosequencing data were ingood agreement with information on community composition generated by high pressure liquid chromatography (HPLC) or clone libraries..". Explain here how HPLC comparison AR: Originally we had submitted a version of this manuscript with detailed information on the methods. The editor had an issue with the description of previously published material and method information. Considering the editors requirements we now cite the relevant publications that contain detailed descriptions of the methodology and comparisions (Kilias et al. 2013 and Metfies et al.2016).

R1: L184 remove e.g if these articles have sufficient information. AR: The sentence was changed according to the reviewer's suggestion (L191).

R1: L186 Why was a nested approach used, this may have implications for the quantification step- how was this overcome? AR: Information was added to clarify this question (L194-197 We used this nested approach, because it minimized the variability between technical replicates of q-PCR data obtained from analyses of field samples. The applicability of the nested approach was evaluated by a comparison of q-PCR data with manual counts of Phaeocystsi pouchetii in field samples (data not shown).)

R1: L195-202 What controls did you use? AR: Information was added to clarify this question (L197-199 In the first step total eukaryotic 18S rDNA was amplified from a positive control ( genomic DNA Phaeocystis pouchetii), a negative control (no template) and genomic DNA isolated from field samples using the universal primer-set. . .).

R1. L203: where does this equation come from? AR: The equation comes from the calibration with a dilution series of a laboratory culture of Phaeocystis pouchetii described in line 201-202 and illustrated in figure 4a.

R1: L235-245 So can it be deployed without any experts on it from start to finish? AR: This is elaborated in line 254-255 of the revised manuscript.

R1: L238- Name the preservative AR: The name of the preservative used in this study is given (L255 Prior to storage, a preservative such as Lysis Buffer RLT (Qiagen, Germany)).

R1: L248 How did you ensure the piping in the ship pump apparatus was clear of microbial biofilm and/or residual water? AR: AUTOFIM is deployed in close proximity to the inflow of the same ships pump system that continuously supplies water to a flow through sensor system (FerryBox) which is installed on RV Polarstern. This fact insures that AUTOFIM does not filter residual water of the ships pump system. The piping of the ship pump system is cleaned in regular intervals to avoid microbial biofilms.

R1: L249 "...at meso- or large in large sample sets". Sentence confusing- meso or large-scale- you mean geographically large or large sample numbers? Re-word if its due to large sample numbers explain why? AR: "at meso- or large" was removed from the sentence.

R1: L258-263 - alter to....marine phytoplankton is CONSIDERABLE - "...dimension is OF particular importance -The word "However" is used but i cannot see a connection between the first sentence and the second. Clarification needed. AR: The sentence was rephrased according to the reviewer's suggestions (L280-282 Identification of pattern in phytoplankton biogeography or biodiversity requires analyses of large samples sets, because spatial heterogeneity of marine phytoplankton is considerable,)

R1: L270: - What do you mean by "deeper horizons"? do you mean greater depths? AR: "deeper horizons" was replaced by "deeper water layers" - I think you need to explain the connection between autofim stations and ctd stations-where they geographically close or was it just depth? AR: It was clarified in the text that the AUTOFIM-filters were collected at the same station/location as CTD-samples (L291). - I would mention that 5m- 50m is within the photic zone. AR: The term "photic zone" was added to the text (L286-287). - Also in what way were they similar- taxonomic assemblage or together with other factors? AR: Additional information on the technical background of ARISA was added at the beginning of the ARISA-paragraph to clarify this question (L268-272 ARISA provides information on variability in protist community structure in larger sample sets at reasonable costs and effort. In an ARISA-analysis the community is characterized by its community profile, which is based on the composition (presence/absence) of differently sized DNA fragments. The DNA fragments are a result of the amplification of the internal transcribed spacer region of the ribosomal operon, which displays a high degree of taxon-related variability in its length).

R1: L271 typo in individual AR: Fixed

R1: L272 ..and WITH the integrated signal FROM THE CTD SAMPLING at all three depths.... AR: The sentence was changed according to the reviewer's suggestions (L293-295 The samples collected with AUTOFIM at stations PS92/19 and PS92/43

clustered together with the individual samples collected at other depth at the same location (5m; 20m; 50m) and with the integrated signal from the CTD sampling all three depths at this location (Figure 3).)

R1: L277: According to THE basin of origin AR: The sentence was changed according to the reviewer's suggestion.

R1: L282- 284: This is extra information so can go. AR: The authors think that this information should stay in the text because it is not redundant and illustrates the quick technological progress in the field.

R1: L284: "...parallel 454-pyrosequencing WAS FOUND TO generate..." AR: The sentence was changed according to the reviewer's suggestion (L307).

R1: L286-7: What sequence data sets? CTD, AUTOFIM or both? AR: Information to clarify this question was added to the text (L311-312 The samples analyzed in the course of this evaluation originated from the same Niskin-bottle of a respective CTD-cast.)

R1: L290: to determine THE variability AR: The sentence was corrected according to the reviewer's suggestion (L314)

R1: L295: Repetitive. Replace "collected in the area of the "Deep-Sea Long-Term Observatory Hausgarten"" with "in that area". AR: The sentence was changed according to the reviewer's suggestion (L319)

R1: L304: Alter to "Development and evaluation of molecular probe based methods: molecular sensors and qPCR" AR: The heading was changed according to the reviewer's suggestion (L328)

R1: L307: "...the surface of the sensor chips THAT BIND to EITHER the rRNA (transcriptome) or rDNA (genome) of the target species." AR: The sentence was corrected according to the reviewer's suggestion.

R1: L306:308: Mention that it is also quantitative and how this is achieved- a diagram of these methods would help readers understand. AR: In line 329 it says already that the method is quantitative, while the detection principle is described and illustrated in the references provided in L331-332.

R1: L308-310: Alter to "Quantitative or real time PCR (qPCR) IS A PCR-BASED METHOD THAT UTLITISES FLUORESCENT DYES OR FLUORESCENTLY-TAGGED DNA PROBES TO QUANTIFY SPECIES BYDETECTING THE AMOUNT OF DNA FORMED AFTER EACH PCR CYCLE". This way the reader can link species abundance with DNA quantity. L310: Also useful for quantifying species. AR: The sentence was re-phrased (L334-336 Quantitative or real time PCR (qPCR) is a PCR-based method that utilizes fluorescent dyes or fluorescently-labelled molecular probes to quantify nucleic after each PCR cycle. Itis a useful tool for quantitation of nucleic acids, respectively species in a given environment)

R1: L314-317: reference needed for this sentence AR: A reference was added (Zhu et al.,2005).

R1: L317: May be use "As such" or another term instead of "In respect to this," AR: "In respect to this" was removed.

R1: L322: What are you measuring"from microscopy, HPLC and flow cytometry". Cell counts, pigments? Add this in. How did you related pigments to cell counts- give a reference to the method? R1: L323: What about the other measurements? AR: The sentence was re-phrased (L347-349 The data on species abundance obtained from of molecular sensors targeting either 18S rDNA or 18S rRNA were evaluated with the results obtained from microscopic counts(Wollschlaeger et al., 2014).)

R1: L324 replace high potential with good potential AR: "high potential" was replaced by "excellent potential"

R1: L325: What do you mean by "the related regular monitoring". Is it qpcr/molecular sensors? If so i suggest Here, additional quantitative molecular monitoring AR: The sentence was changed (L352-354 Here, the regular quantitative molecular monitoring would benefit from advantages like reduced effort, and the high potential for automation of the methodology (Wollschlaeger et al., 2014))

R1: L326: reduced effort in what? Change high potential to excellent or good potential. AR: The effort was specified (L353 (time, costs and labor))

R1: L333: delete while AR:"While" was deleted

R1: L335:PSU, than.... Delete comma. AR: Comma was deleted

R1: L343 reference needed for the 2014 findings AR: The 2014 findings are presented in figure 4 of this manuscript.

R1: L345: This study also suggested this positive correlation. Suggest This study also found a positive correlation in agreement with XYZ, et al 2014. AR: The sentence was re-phrased (This study also found a positive correlation in agreement with the findings of 2014, even though sequence abundance of Phaeocystis pouchetii was more evenly distributed in Fram Strait in 2012(Metfies et al., 2016))

R1: L352: hierarchically organized molecular based. I would add that its a combined autonomous sampling and molecular testing platform. AR: The information was added (L379-380 Here we introduce for the first time an integrated hierarchically organized molecular based observation strategy that combines autonomous sampling with molecular analyses.)

R1: L360 change strong to excellent/good AR: "strong" was changed to "excellent"

R1: Figures I think map figure would be really helpful to allow readers to understand spatial Comparisons AR: This manuscript reviews the findings of ∼ 10 publications, that all contain maps of the respective research area, references for the maps are provided. Figure 4 contains a map for the newly published data on Phaeocystis pouchetii abundance in Fram Strait in summer 2014.

R1: Also for section 3.1.4 for readers who are not familiar with molecular methods a diagram of how a qpcr/molecular sensors work or a photograph of the one you have would be good- you could alter fig 1 as its quite small and provide a clearer picture of these? AR: Figure 1 was revised and contains now diagrams that explain the background of the analyses used in the observation strategy.

R1: Fig. 2: Would be good to see basic diagram of its layout and its connected and its modules. AR: The layout and technical details of the device is not in focus of this manuscript. These informations will be published elsewhere.

R1: Fig. 3: Define Meta MDS. Methodology needs to be referred to in methods. I would explain the labelling system for expeditions and stations. Where/when were the other PS stations and what did they represent? AR: The figure legend was extended to provide this information. The method used to generate the metaMDS plot is described in Kilias et al. which is cited in the material and methods section.

R1: Fig. 4: A- i would explain the significance of that graph to non-expert readers, to say the assay worked and provided a good relationship between DNA quantity and cell abundance. B- What do the numbers next to the points in the map represent? C: parameter needs to be plural,. What does the inset graph show? What do the numbers represent by the triangles? I suggest explain the plot. AR: The figure legend was rephrased and more elaborate information to answer the questions of reviewer 1 were added. (Assessment of Phaeocystis pouchetii in Fram Strait. A: Calibration of Phaeocystis pouchetii specific qPCR assay with a dilution series of laboratory cultures. The CT value is significantly correlated with cell numbers. B: Abundance of Phaeocystis pouchetii in Fram Strait. The dots and the associated numbers represent sampling sites and associated station numbers of expedition ARKXXVIII(PS85) of RV Polarstern in summer 2014, while cell numbers/liter are reflected by different colours. C: Principal component analysis including environmental parameters (temperature, salinity, Chl a biomass and sea ice coverage) and abundance of Phaeocystis pouchetii. Triangles and associated numbers represent sampling sites and associated station numbers of expedition ARKXXVIII(PS85) of RV Polarstern in summer 2014. HG4 indicates the central station of the "Deep-Sea Long Term Observatory Hausgarten" in Fram Strait. The Eigenvalues indicate the proportion of variance explained by different dimensions in the diagram. The black bars in the histogram reflect the x-axis and the y-axis. Here ∼ 80% of variance is explained in this two-dimensional diagram of the PCA (x-axis: 50.29%; y-axis: 30.08%).)

Please also note the supplement to this comment:
http://www.ocean-sci-discuss.net/os-2016-23/os-2016-23-AC1-supplement.pdf

[Figure]

**Supplement:**

[revised manuscript text omitted]

---

## Author Comment (AC2) · 30 Aug 2016

Author's Reply (AR) to Anonymous Referee # 2 (R2) R2: A bit more thorough overview of related technology could be relevant, however. For instance the ESP (Environmental Sample Processor) has an automated filtration unit directly connected to the possibility of using qPCR or hybridization for microbial species identification under water (although not as part of a ferry box system as far as I know). AR: L122 The ESP is cited in the manuscript in line 120 (Ussler et al., 2013).

R2: Firstly, the authors should explain what they mean by "underlying water column".

They did not present vertical CTD profiles of the water column, so it is difficult to know if the samples taken from distinct depths using the Niskin bottles were all taken from the same layer. AR: L286-287 This sentence was rephrased (In respect to this it was necessary to evaluate how representative samples from 10 m depth might be for the photic zone in the underlying water column.)

R2: Their results show that at the (only) two stations sampled for the comparison, the AUTOFIM sample communities at 10m were associated with the communities collected using Niskin bottles from the same water layer. Their discussion around this result (around lines 270-274) is a bit unclear referring how the AUTOFIM samples is representative of that of "deeper horizons". What do the authors mean by this? I assume they mean that the AUTOFIM samples are representative for the upper mixed layer, because they do not have data from deeper layers. This part should be rephrased/clearified. AR: L291-295 This part was rephrased (The ARISA patterns obtained from deeper water layers of the photic zone (20m; 50m) are highly similar to those obtained from the samples collected with AUTOFIM. The samples collected with AUTOFIM at stations PS92/19 and PS92/43 clustered together with the individual samples collected at other depth at the same location (5m; 20m; 50m) and with the integrated signal from the CTD sampling all three depths at this location.)

R2: The automated biosensor system used - it is a bit unclear to me exactly what this is. Do they refer to ferry box data or to analyses performed on the filtered seawater collected using the AUTOFIM? If the latter, which sensors/analyses do they refer to? Or do the refer to the molecular sensors of Wollschlaeger et al. (2014)? AR: Yes, we are referring to the molecular sensor experiment described and evaluated in Wollschlaeger et al., 2014.

R2: In the introduction, when the authors refer to the molecular tools available (line 105 and onwards), they mainly refer to their own work. But there are several studies that would be relevant to include in such an overview. I suggest the authors refer also to other studies from Arctic waters, in particular the Canadian Arctic has been

explored using similar and relevant molecular tools. AR: L114 In this manuscript we cited Sunagawa et al., 2015, which is one of the most relevant manuscripts published in the field during the past two years. In addition to this we complemented our citations with Comeau et al., 2011, a description of the Arctic microbial community structure before and after the record sea ice minimum in 2007.

R2: line 152: Were the particles for molecular analyses added a buffer? Or perhaps stored in -80 prior to DNA extraction? AR: L155-156 Information to clarify this was added to the text (Subsequent to sampling filters were stored at -20°C until further analyses.

R2: line 165: Is the use of the E.Z.N.A DNA extraction kit correct? And was it used for both the AUTOFIM and Niskin bottle samples? If so, why was the the Qiagen lysis buffer added to the filters collected using AUTOFIM? AR: Yes, the E.Z.N.A. DNA extraction kit is correct and it was used for both the AUTOFIM and Niskin bottle samples. We used the Qiagen lysis buffer, because it is on one hand exchangeable with the E.Z.N.A. lysisbuffer and on the other hand preservation with QLT-buffer left the possibility to use the AUTOFIM-samples for quantification with our automated biosensor system.

R2: line 178: ITS1 is the internal transcribed spacer 1. It is also an "intergenic spacer region", but the use of that term without explaining the ITS1 abbreviation is a bit confusing. AR: L185 "intergenic spacer region" was replaced by "internal transcribed spacer"

R2: first paragraph of 3.1 is mostly repeating what is already pointed out in the introduction. This section could be reduced. AR: Paragraph 3.1 describes the observation strategy that is not mentioned in the introduction. Thus it did not get clear to the authors, which parts of the paragraph reviewer 2 is referring to.

R2: line 259: Rephrase sentence, the word "scale" lacking? line 261: "from" should be replaced with "of" (particular importance). AR: L 278-284 This sentence was rephrased (We suggest to use ARISA as part of the molecular observation strategy to identify biogeographic or biodiversity patterns in large sample sets, e.g. collected via AUTOFIM.

Identification of pattern in phytoplankton biogeography or biodiversity requires analyses of large samples sets, because spatial heterogeneity of marine phytoplankton is considerable, while the vertical dimension is of particular importance,….)

R2: line 297-299, incl Metfies et al 2016: Is the % cells of Phaeocystis due to % reads or quantitative counts? If it refers to % reads, the statement is a bit strong. AR: L320-323 The statement is based on measurements of Chl a biomass subsequent to fractionated filtration. The sentence was rephrased to clarify the uncertainty (A larger survey of Arctic protist community composition in 2012 including Fram Strait and larger parts of the Central Arctic Ocean confirmed these observations and identified Phaecystis pouchetii again as an important contributor to Arctic pico-eukaryote Chl a biomass. The latter constituted between 60-90% of Chl a biomass during summer 2012 in the Arctic Ocean (Metfies et al., 2016).)

R2: Fig. 1 text: This text should explain the different levels of the observation strategy in greater detail, so that it is not necessary to check the manuscript text to identify the different parts. AR: Information on the different levels of the observation strategy was added (Overview of the smart observation strategy which is organized in four different levels: level 1: samples are collected underway or at monitoring sites using the remote-controlled automated filtration system AUTOFIM; level 2: direct molecular surveillance of key species aboard the ship via an automated biosensor system or quantitative polymerase chain reaction; level 3:.preserved samples are analyzed via molecular fingerprinting methods (e.g. ARISA) that provide a quick and reliable overview of differences in protist community composition of the samples in a given observation area or time period; level 2: detailed analysis of taxonomic protist composition in selected samples via latest next generation sequencing.

R2: Fig. 3 text: Samples collected via CTD ... imprecise, the samples were collected using Niskin bottles. AR: The sentence was rephrased (Fig. 3: MetaMDS Plot (non metric multidimensional scaling plot) of ARISA fingerprints generated from samples collected via Niskin bottles coupled to a CTD-rosette and AUTOFIM.The closer

the samples are located to each other in the metaMDS-plot, the more similar are the ARISA-profiles of the samples. The label of the samples gives information on the cruise leg (PSXX) and the station (/XX). Samples were collected during expeditions PS92 and PS 94 of RV Polarstern to the Arctic Ocean during summer 2015. The samples collected during PS94 serve as an outgroup in this analysis. )

R2: Fig. 4 text: This text also does not explain the figure very well. Do the numbers represent station numbers? The eigenvalues histogram in 4C is not explained - what do the black vs white histograms signify? What values are at the y axis? AR: The figure legend was rephrased (Assessment of Phaeocystis pouchetii in Fram Strait. A: Calibration of Phaeocystis pouchetii specific qPCR assay with a dilution series of laboratory cultures. The CT value is significantly correlated with cell numbers. B: Abundance of Phaeocystis pouchetii in Fram Strait. The dots and the associated numbers represent sampling sites and associated station numbers of expedition ARKXXVIII(PS85) of RV Polarstern in summer 2014, while cell numbers/liter are reflected by different colours. C: Principal component analysis including environmental parameters (temperature, salinity, Chl a biomass and sea ice coverage) and abundance of Phaeocystis pouchetii. Triangles and associated numbers represent sampling sites and associated station numbers of expedition ARKXXVIII(PS85) of RV Polarstern in summer 2014. HG4 indicates the central station of the "Deep-Sea Long Term Observatory Hausgarten" in Fram Strait. The Eigenvalues indicate the proportion of variance explained by different dimensions in the diagram. The black bars in the histogram reflect the x-axis and the y-axis. Here $\sim$ 80% of variance is explained in this two-dimensional diagram of the PCA (x-axis: 50.29%; y-axis: 30.08%). )

Please also note the supplement to this comment:
http://www.ocean-sci-discuss.net/os-2016-23/os-2016-23-AC2-supplement.pdf

―――――――――――――――――